# Statistical Approach to Assess Chill and Heat Requirements of Olive Tree Based on Flowering Date and Temperatures Data: Towards Selection of Adapted Cultivars to Global Warming

Omar Abou-Saaid [1,2,3], Adnane El Yaacoubi [4], Abdelmajid Moukhli [1], Ahmed El Bakkali [5], Sara Oulbi [1], Magalie Delalande [2], Isabelle Farrera [2], Jean-Jacques Kelner [2], Sylvia Lochon-Menseau [6], Cherkaoui El Modafar [3], Hayat Zaher [1] and Bouchaib Khadari [2,7,*]

1   INRA, UR Amélioration des Plantes, Marrakech 40005, Morocco
2   AGAP Institut, Université de Montpellier, CIRAD, INRA, Institut Agro, 34398 Montpellier, France
3   Centre d'Agrobiotechnologie et Bioingénierie, URL-CNRST 05, Université Cadi Ayyad, Marrakech 40010, Morocco
4   Ecole Supérieur de Technologie, Université Sultan Moulay Slimane, Khenifra 54006, Morocco
5   INRA, UR Amélioration des Plantes et Conservation des Ressources Phytogénétiques, Meknes 50102, Morocco
6   Conservatoire Botanique National Méditerranéen de Porquerolles (CBNMed), 83400 Hyères, France
7   CBNMed, AGAP Institute, 34398 Montpellier, France
*   Correspondence: bouchaib.khadari@cirad.fr or b.khadari@cbnmed.fr

**Abstract:** Delineating chilling and forcing periods is one of the challenging topics in understanding how temperatures drive the timing of budburst and bloom in fruit tree species. Here, we investigated this question on olive trees, using flowering data collected over six years on 331 cultivars in the worldwide collection of Marrakech, Morocco. Using a Partial Least Squares approach on a long-term phenology (29 years) of 'Picholine Marocaine' cultivar, we showed that the relevance of delineating the chilling and forcing periods depends more on the variability of inter-annual temperatures than on the long-term datasets. In fact, chilling and forcing periods are similar between those delineated by using datasets of 29 years and those of only 6 years (2014–2019). We demonstrated that the variability of inter-annual temperatures is the main factor explaining this pattern. We then used the datasets of six years to assess the chill and heat requirements of 285 cultivars. We classified Mediterranean olive cultivars into four groups according to their chill requirements. Our results, using the Kriging interpolation method, indicated that flowering dates of most of these cultivars (92%) were governed by both chilling and forcing temperatures. Our investigations provided first insights to select adapted cultivars to global warming.

**Keywords:** *Olea europaea* L.; flowering data; partial least squares regression; dynamic model; chill requirements; climate change; Mediterranean fruit tree; adapted cultivars

## 1. Introduction

The olive tree constitutes a remarkable species by its biological and ecological characteristics widely cultivated in many regions of the world, particularly in the Mediterranean area [1]. However, this crop is facing climatic constraints in the current context of global warming, perturbing its biological, physiological and phenological development [2]. Air temperature is the most important abiotic factor affecting olive development. It is mainly involved in the dormancy and flowering process during winter and spring, respectively [3,4]. The bud dormancy onset and its breaking date phase are strongly influenced by winter chill; meanwhile, the flowering achievement is highly correlated to spring heat [5]. In addition to the biennial bearing of the olive, the annual temperature variations during these two periods seem to have significant negative consequence on the development cycle of tree production [4–6], resulting in economic repercussions [7]. At a phenological level, it was reported that an increase in temperature during winter and spring induced the flowering

advance of olive cultivars in some Mediterranean areas such as Morocco, France [4], Spain, Italy and Tunisia [8,9]. Furthermore, seasons with warm temperatures during winter and spring could delay the chilling period, making more time for trees to accumulate their chill requirements. On the other hand, this increases the bud differentiation, accelerating the flowering achievement and making the heat period shorter because of the fast satisfaction of heat requirements induced by warm spring temperatures [10–12]. However, the delay of the chilling period was somewhat recompensed by the heat period shortening, which explains the flowering advance reported in many previous investigations [13–16].

The satisfaction of olive chill requirements remains necessary. It was reported that high temperatures (above 16 °C) during the floral initiation prevented the development of flower buds, inducing an abnormal reproductive cycle [17]. In favorable conditions, Lavee et al. [18] highlighted that exposure of the olive tree to low temperatures during the winter appears to be a substantial element in producing normal bud induction, initiation, and floral differentiation. This study was confirmed later, showing that the rate of flower bud differentiation was higher during colder seasons than warmer ones [19]. Physiologically, this result was explained by the inhibitory effect of high temperatures on the induction and subsequently on the flower development [20,21]. Understanding the effect of chill and heat requirements on the reproductive cycle presents some complexity, given that winter chill and spring heat accumulation assessment (time of bud dormancy onset, duration of the endo/ecodormancy periods, bud break, flowering achievement, start/end of each phenological stage, etc.) require hard and long experimentation, making its realization very difficult, particularly when the number of the studied sites and cultivars is very high (more than 100 cultivars). Recently, new statistical approaches showed efficient results in terms of identification of the endodormancy and the ecodormancy periods and determination of tree chill and heat requirements. The Partial Least Squares (PLS) regression is one of the most efficient tools used in several studies on forest and fruit tree species, such as apple, almond [22–24], cherry [25], walnut [26], and chestnut [27]. It requires simple climatic and phenological data (e.g., daily temperatures and annual flowering dates) recorded and collected over several years. Afterwards, the use of such a statistical approach could easily facilitate classification of cultivars according to the relevant duration of their phenological phases and chill/heat requirements.

In this investigation, we aimed to use the PLS approach to determine these important unseen phenological periods (chilling and forcing phases) and consequently to identify the date of breaking dormancy involved in the flowering achievement of olive. The determination of these periods will lead us to estimate precisely the chill and heat requirements of a subset of olive cultivars cultivated in the World Olive Germplasm Bank of Marrakech, Morocco (WOGBM), consisting of 285 cultivars, in order to understand the responses of olive phenology events to temperature variation during these relevant periods.

## 2. Materials and Methods

### 2.1. Study Site and Plant Material

Phenological observations were carried out in the WOGBM at the INRA research experimental station of Tassaout, about 60 km northeast of Marrakech, Morocco. The WOGBM is located at 31°49′10″ N; 7°25′58″ W (CRS: WGS84-EPSG:4326); 465 m of altitude, under semi-arid climate conditions, with an average of annual mean temperature (calculated during the period 1972–2019) of 18.84 °C ranging from 16.62 °C to 20.25 °C (Figure S1 (Supplementary Materials)); the soil is characterized by a clay loam texture.

Phenological observations were carried out on 331 cultivated *Olea europaea* (L.) cultivars identified based on 554 accessions and originating from 14 Mediterranean countries [28,29]. These accessions were previously characterized using 20 SSR markers and 11 endocarp traits [30]. Each cultivar is represented by at least 3 trees. As synonymy, and redundancies cases were revealed, some cultivars such as 'Beladi' and 'Picholine Marocaine' consisted of 29 and 20 accessions, respectively. These accessions were represented by 106 trees for 'Beladi' and 88 trees for 'Picholine Marocaine' (Table S1). Olive accessions were

progressively introduced in the WOGBM collection between 2003 and 2012. Trees were planted at $7 \times 4$ m and grown under the same cultural practices, drip irrigation system, and the same protocol of pruning and fertilizing.

### 2.2. Climate and Phenological Records

According to the BBCH scale [31], phenological stages related to the olive inflorescence emergence and flowering were recorded over six years, 2014–2019, over all of the WOGBM. Observations were carried out every 2 or 3 days from the first of February to the end of the flowering period to determine the date of inflorescence emergence stages (stage 51 to stage 59) and flowering stages (stages 61, 65, and 69) according to the BBCH scale for olive trees [31]. Additionally, flowering dates' data of 'Picholine Marocaine' cultivar, located in the same study site for 23 years of full flowering date (FFD) data from 1986 to 2013, were kindly provided by El Yaacoubi et al. [4]; however, the FFD data between 1992 and 1996 were not available.

Hourly temperatures' data from 2014 to 2019 were provided from the weather station of Tassaout, where the WOGBM cultivars are grown. Data were collected through a website connected to the weather station (FieldClimate) operated by Pessl Instruments GmbH (Austria), with some gaps due to missing recorded data. Therefore, these gaps were filled by data generated from recorded daily maximum and minimum temperatures of the study site, using the chillR package in R software [32]. Accordingly, for the period from 1972 to 2012, hourly temperature data were generated based on daily maximum and minimum temperature data provided by El Yaacoubi et al. [4] using the chillR package. ChillR implements the two following equations [33]:

$$T(t) = (T_{max} - T_{min}) \times \sin [(\pi \times t)/(DL + 4)] + T_{min} \tag{1}$$

where $T(t)$ is temperature at time t after sunrise; $T_{max}$ is maximum temperature; $T_{min}$ is morning minimum temperature; and DL is daylength (in hours):

$$T(t) = Ts - [(Ts - T_{min})/(24 - DL)] \times \ln(t) \tag{2}$$

where $T(t)$ is temperature at time $t > 1$ h after sunset and Ts is the sunset temperature obtained from Equation (1).

Differences in daylength between locations are accounted by computing sunrise and sunset times based on geographic latitude using chillR package [32].

All statistical data analyses were run in the R programming environment (R Development Core and Team, 2021; version R 3.6.3, Vienna, Austria) [34].

### 2.3. Variability Analysis of Flowering Dates

Phenological data have been converted according to their corresponding Julian days, starting from the first of January of each year (Day of the Year (DOY)). Correlations of phenological stages recorded as Julian days for all trees per cultivar for all years were investigated by Pearson correlation analysis using the Performance Analytics package [35].

FFD (corresponding to the Julian date of the stage 65) recorded on the 331 cultivars of the WOGBM over the six years were firstly analyzed using an Analysis of Variance (ANOVA) to investigate "cultivar", "year" and the interaction "cultivar $\times$ year" effects considering the cultivar variance for each year. Mean comparison of full flowering dates for the 331 cultivars was carried out with Tukey's test at the significance level of 5%.

Following the first classification of the 331 studied cultivars according to their FFD based on ANOVA and Tukey's test, data were then investigated by the linear mixed model using the Lme4 package [36] to assess the effect of the cultivar, the year, and the "cultivar $\times$ year" interaction factors. Three mixed-effect models were tested and compared: (i) the model-1 with only the cultivar as a random effect; (ii) the model-2 with the cultivar as a random effect and the year as a fixed effect and (iii) the model-3 called the full model with interaction "cultivar $\times$ year" as a second random effect.

Full model equation:

$$Y = \mu + Gi + Aj + (GA)ij + \varepsilon$$

where Y is the phenotypic value measured on k trees of cultivar i in year j; $\mu$ is the overall mean; Gi is the random effect of the cultivar i; Aj is the fixed effect of year j and (GA) ij is the random interaction between cultivar i and year j, while $\varepsilon$ is the random residual error.

The best model was selected based on the Akaike Information Criterion and Bayesian Information Criterion (AIC and BIC; Table 1). Variance components were then used to estimate the broad-sense heritability ($\mathbf{H^2}$) as:

$$\mathbf{H^2} = \frac{\sigma_G^2}{\sigma_G^2 + \sigma_{\mathbf{GxA}/J}^2 + \frac{\varepsilon^2}{n}}$$

where $\sigma_G^2$ is the variance of cultivar effect; $\sigma_{\mathbf{GxA}/J}^2$ is the variance of "cultivar × year" interaction effect; $\varepsilon^2$ is the variance of the residual term; $J$ is the number of years and $n$ is the total number of observations.

**Table 1.** Linear mixed analysis demonstrating the relationship between full flowering date (FFD) and cultivars over years. $X^2$ is a Chi squared test from the comparison of models, Alpha is 0.05 and Df is the degree of freedom. AIC and BIC are two criterions of the best model, as the value of the criterion decreases the model fits the data better. $H^2$ is the heritability in the broad sense calculated from the model. Cultivar × Year is the interaction between cultivar and year effects. Levels of significance: ns (not significant); *** ($p < 0.001$).

| Mixed Model | Equation | Df | AIC | BIC | $X^2$ | $p$ Value | $H^2$ |
|---|---|---|---|---|---|---|---|
| Model-1 | FFD = ǀCultivar | | 44,967 | 44,987 | | | 0.098 |
| Model-2 | FFD = Year + ǀCultivar | 5 | 31,049 | 31,102 | 13,928 | <0.001 *** | 0.79 |
| Model-3 | FFD = Year + ǀCultivar + ǀCultivar $^2$ × Year | 1 | 29,639 | 29,699 | 1412 | <0.001 *** | 0.75 |

The best linear unbiased prediction (BLUP) was extracted from the best-mixed model to assess the genetic effect of cultivars on the full flowering trait. Cultivars were then classified according to three criteria: (i) based on full flowering date as DOY; (ii) based on the best linear unbiased prediction (BLUP) assessed for the full flowering date and (iii) based on chilling requirements of each cultivar. A method implemented in NbClust package in R was used to choose the best number of clusters [37]. The method provides arguments to specify the dissimilarity distance between observations and the agglomeration method to use in clustering. Here, we chose Euclidean distance for dissimilarity measures and Ward's method for hierarchical clustering.

### 2.4. Temperature Trends

Based on annual mean, maximum and minimum temperatures recorded from the weather station of Tassaout during the period 1972–2019, trends were analyzed by linear regression to test the temperature increase tendency linked to global warming using ggplot2 and ggpubr packages [38,39]. Furthermore, to assess temperature variations according to the seasons, the generated dataset of hourly temperature during the period from 1972 to 2012 and the registered dataset from 2013 to 2019 were analyzed using the chillR package.

### 2.5. Delineating Chilling and Forcing Phases Based on PLS Analysis

The purpose of PLS is to find the aspects of the signal in one matrix (e.g., flowering date) that are related to signals in another matrix (e.g., temperatures). For the purpose of this study, we focused on the independent variables; daily mean temperatures for the entire year and over the years of flowering data records whereas the dependent variables were the observed FFD, expressed in Julian days. Daily mean temperatures are replaced by an 11−day running mean starting 5 days before and 5 days after the corresponding date. Here,

two analysis outputs are presented: (i) the variable importance plot, which represents the contribution of each predictor in fitting the PLS model for both predictors and response and (ii) the regression coefficient profile, which shows the direction and strength of the impact of each variable in the PLS model. The variable importance plot shows the individual predictor variables (x-variables) and the corresponding scores of the Variable Importance in the Projection (VIP) statistic [40]. The VIP scores are based on a weighted sum of squares of the PLS loadings and calculated for each variable. For interpretation purposes, only predictors with a VIP of more than 0.8 are considered significant [40]. The regression coefficient profile is obtained by plotting the model coefficients of the centered and scaled data against the predictor variables. Centering and scaling of dependent and independent variables is necessary to allow comparison between different variables, with respect to their influence in the model. Predictor variables with high VIP scores and high absolute values for model coefficients represent therefore the periods in which temperature changes will have the strongest effect on the FFD.

The chilling and forcing phases of the studied cultivars were identified with regards to the model coefficients and the, at least temporarily, important VIP scores. Positive coefficients imply that the temperatures on these days were positively correlated with the flowering dates, i.e., lower temperatures during winter lead to earlier flowering dates (equivalent to the chilling accumulation period). In contrast, negative coefficients signify that the temperatures were negatively correlated with the flowering dates, i.e., higher temperatures in spring also contribute to earlier flowering (equivalent to the forcing period) [9]. This is consistent with the notion of chill requirements, according to which warm temperatures should delay the dormancy breaking and thus lead to later flowering.

Following this approach, the PLS analysis was used to delineate chilling and forcing of each of the studied cultivars (Table S1). Due to missing data, only 285 cultivars were used for the PLS analysis (Table S1).

### 2.6. Comparison of Chilling Models

The most widely used chill models in horticulture are the Chilling Hours model (CHM) [41,42], the Utah model (UM) [43] and the Dynamic model (DM) [44,45]. To identify the most accurate model for our data and geographic area, we used the long-term phenological records of 'Picholine Marocaine' cultivar data for 23 years, from 1986 to 2013 [4] and those of 6 years, from 2014 to 2019, collected for the present study. With respect to chilling models, the variation of accumulated winter chill during the distinct phases also provided an indication of model quality. As we hypothesize that the chill requirements are predetermined genetically, we can expect the amount of winter chill that was accumulated in the time window to be similar across years. We therefore calculated the coefficient of variation in the chill estimates for the period delineated by PLS approach for all three models. The model showing the lowest coefficient of variation was considered as the most accurate among the tested models.

### 2.7. Chill and Heat Computation

Chill and heat accumulation were estimated using the Dynamic Model [45] and the Growing Degree Hour model (GDH) [46], respectively. Once chilling and forcing phases were delineated for each of the studied cultivars by the PLS analysis, the calculation of chill and heat accumulation from the initial date to the end date of each phase for each year was calculated using the chillR function 'chilling'. Means of chill and heat requirements over the six years and its standard deviation were taken as the estimation of the average chill requirements and heat requirements for each of the cultivars (Table S8). Similarly, groups of cultivars according to their CR and HR estimated values were clustered according to the method of Charrad et al. [37].

*2.8. Impacts of Chilling and Forcing Temperatures on Flowering Dates*

To better understand which temperatures have a greater effect on the flowering dates, chilling or heat temperatures, we plotted FFD against the mean temperatures during the whole chilling and forcing periods (revealed by PLS) using the Kriging interpolation method [47,48]. The color spectrum of the plots must be interpreted as advancements or delays in the flowering date. The isolines were created to represent homogeneous flowering date. Closer points tend to have more similar values than the distant ones.

## 3. Results

*3.1. Temperature Trends and Chill Availability*

The average of min and max temperatures increased about 0.28 °C and 0.41 °C each decade, respectively, potentially showing global warming (Figure S2). The rainfall average during the same period 1972–2019 was 257.08 ± 94.12 mm with a high interannual variation, since the lowest value was 29.5 mm in 1983 and the highest one was 468.8 mm in 2018 (Figure S3). Despite this variability, there is no trend of annual rainfall decreasing during these last five decades (Figure S3).

Computation of chill accumulation in our study site during the period between 1972 and 2019 using the three models, CHM, UM and DM, showed that the accumulation of chilling starts at the beginning of November (Figure S6).

*3.2. Flowering Trends, Variability and Cultivar Classification*

Based on statistical analysis, our results showed that all phenological stages were correlated. High correlations were observed within the inflorescence emergence stages (Stage 51, 54, and 55), and flowering stages (Stage 60, 61, 65, and 69), with values ranging for emergence stages between 0.80 and 0.88, respectively, and for flowering stages between 0.90 and 0.98, respectively. A low correlation between stage 51 and flowering stages was observed (Pearson correlation ranged between 0.024 to 0.45) (Figure S4). A significant negative correlation was observed between the flowering duration and the inflorescence emergence stages (correlation value ranged between −0.41 and −0.58).

The variance analysis showed an important significant year effect, followed by cultivar, as well as the interaction effect on FFD (Table S2, Figure S5). Full flowering dates for olive cultivars ranged over years between 91 DOY (1 April) in 2019 to 150 DOY (30 May) in 2016. Consequently, over the six years, the range of flowering dates' variation was 59 days. Whatever the considered flowering cultivar group, we observed a considerable standard deviation for mean values highlighting the importance of interannual variability of the flowering date (Figure S5). Tukey analysis revealed the full flowering groups as F1 to F21.The median between the earliest flowering group (F1) and the latest one (F21) was ranged between 113 (23 April) and 137 (17 May; Figure 1 and Table S3). Clustering of cultivars according to Ward's method showed four major groups of flowering dates: the early, mid, late and extra late flowering groups, with an average of flowering dates of DOY 119 (29 April), 123 (3 May), 125 (5 May), 128 (8 May), respectively (Tables S4 and S5).

To consider the year and cultivar interaction effect (cultivar × year), we performed a linear mixed analysis of cultivars' full flowering dates observed in the WOGBM by using and comparing three models (Table 1). We noted a high significant effect of the year as ($X^2(2) = 13928$, $p < 0.001$) and the interaction cultivar × year as ($X^2(3) = 1412$, $p < 0.001$). The broad sense heritability $H^2$ calculated from the model-3 with the lowest BIC and AIC criteria showed a value of 0.75.

The best linear unbiased prediction (BLUP) was calculated from the mixed model-3 to extract the cultivar effect on the full flowering date, which was ranged between DOY 110 (20 April) to DOY 122 (2 May). Similarly, according to their BLUP of the full flowering date, the cultivars were clustered into four groups (Figure 2; Table S6) ranging from the early flowering cultivar group with an average value of DOY 114 (24 April) to the extra late flowering cultivar group with an average value of DOY 120 (30 April).

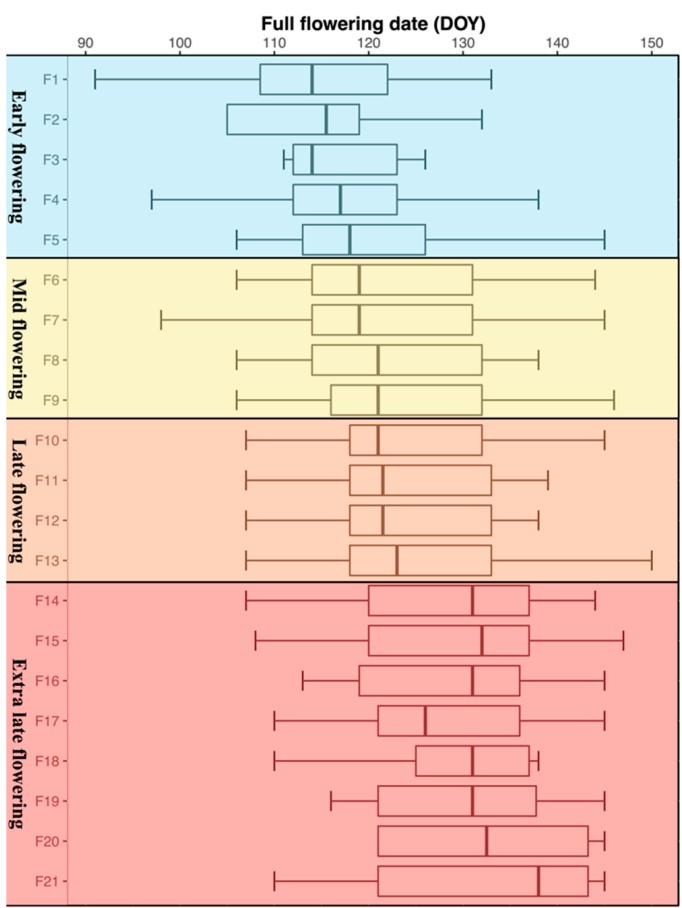

**Figure 1.** Classification of the full flowering date records DOY "day of the year" of the 331 olive cultivars observed from 2014 to 2019. The first and third quartiles are represented by each box's left and right sides, respectively, and the band inside the box is always the median. The whiskers' tips signify standard deviations above and below the mean. Olive tree cultivars were classified into four significant groups: early ▉, mid ▉, late ▉ and extra late ▉ flowering cultivars.

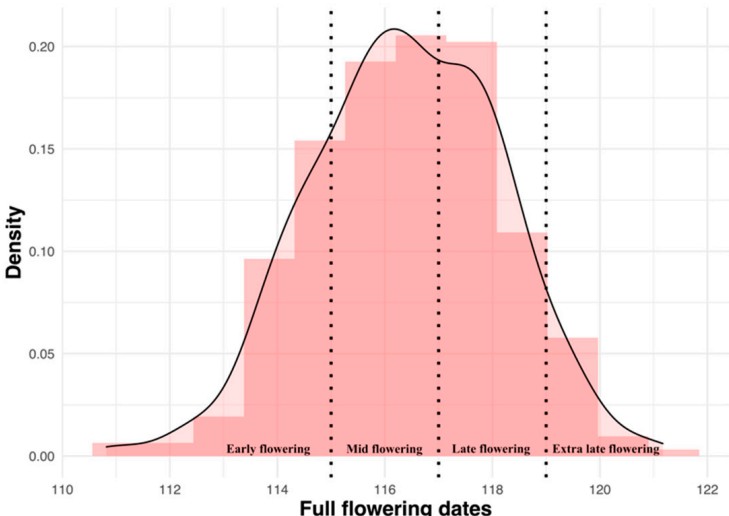

**Figure 2.** Distribution of the best linear unbiased predictors (BLUPs) of flowering date cultivar values extracted from the model-3. The dotted vertical lines are limiting full flowering cultivar groups as defined by the Ward clustering analysis (see Table S6).

### 3.3. Delineating the Chilling and the Forcing Periods of 'Picholine Marocaine' Cultivar

Based on the long-term flowering data of 29 years (1986–2019) recorded for the 'Picholine Marocaine' cultivar, PLS analysis was conducted to: (i) determine the effect of the main temperatures on the full flowering dates and (ii) to identify precisely the critical phases of chill and forcing periods necessary for the olive flowering process. Based on the chill availability assessment using the Chilling Hours model, the Utah model and the Dynamic model during the period from 1972 to 2019, which was occurring during 152 days on average starting generally from the first week of November until the end of March and the beginning of April (see Figure S6a,b), we chose to remove all periods, identified by consistently positive and at least temporarily important (according to the VIP score) model coefficients, out of the defined chill availability phase. Accordingly, the chilling phase of 'Picholine Marocaine' cultivar in the experimental station of Tassaout, Morocco, could be delineated from 27 December to 23 February. Strikingly, we noted a brief period of 15 days of stable negative coefficients from 25 January to 10 February during which this period presented a good fitting with chill availability phase previously defined by the three models (see Figure S6a,b). We therefore considered the chilling phase of 'Picholine Marocaine' starting from 27 December to 23 February (Figure 3).

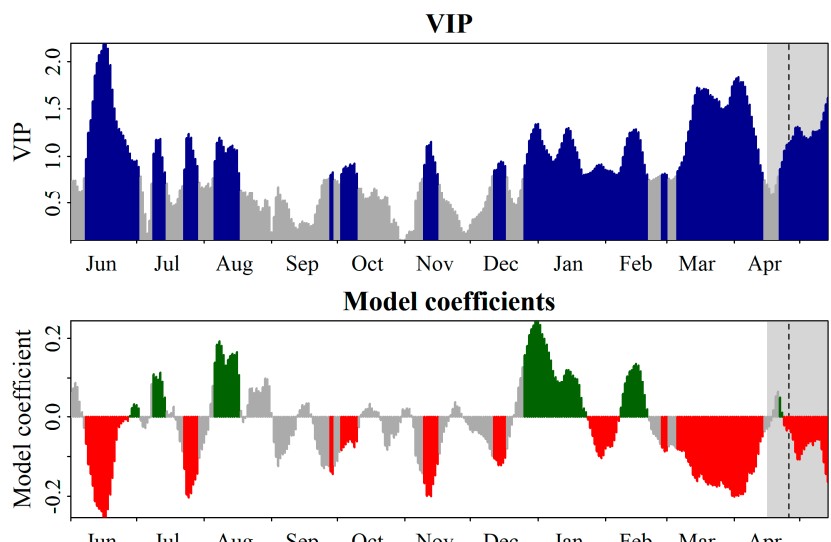

**Figure 3.** Partial Least Squares (PLS) regression of full flowering dates for 'Picholine Marocaine' cultivar recorded over 29 years (1986–2019; FFD data were not available between 1992 and 1996) in the INRA−Tassaout experimental station (Morocco), using 11−day running means of daily mean temperatures. The top panel and the bottom panel correspond, respectively, to the Variable Important in the Projection (VIP) and the Model coefficient of the centered and scaled data. Blue bars in the top panel indicate values above 0.8, the threshold for variable importance indicating that temperatures in these periods affect the full flowering date and gray areas where temperatures have no effect. In the bottom plot, data are shown in red whenever model coefficients are negative, and when they are green, they are positive. The gray area and the dashed line at the right of the two plots indicate, respectively, the variation and the median of flowering date over the 29 years.

The model coefficients showed negative values from late January to mid-May, except for the two short periods, from 10 February to 23 February (13 days) and from 21 April to 23 April (3 days), respectively. The first brief period of 13 days was considered as part of the chilling accumulation phase, while the second, of 3 days, was observed as no expected results, in the same way as the brief period of 15 days of stable negative coefficients, from 25 January to 10 February. We thus considered the heat phase for 'Picholine Marocaine' cultivar as delineated from 24 February to the flowering date averaged to 16 May (Figure 3).

### 3.4. PLS Comparison between Long- and Short-Term Flowering Date Records

Following the delineating chilling and forcing phases of 'Picholine Marocaine' cultivar based on the long-term FFD recorded over 29 years (1986 to 2019; Figure 4 top panel), we compared the PLS outputs using a short-term record data (6 years from 2014 to 2019, Figure 4 bottom panel) corresponding to the period of phenology records of the WOGBM. As mentioned previously in the Figure 3, periods with significant effect (VIP > 0.8) and positive or negative model coefficients out of the defined period of chill availability by chilling models used in this study were removed from chilling and forcing delineation. Interestingly, we noted a similar pattern between the two series (Table 2).

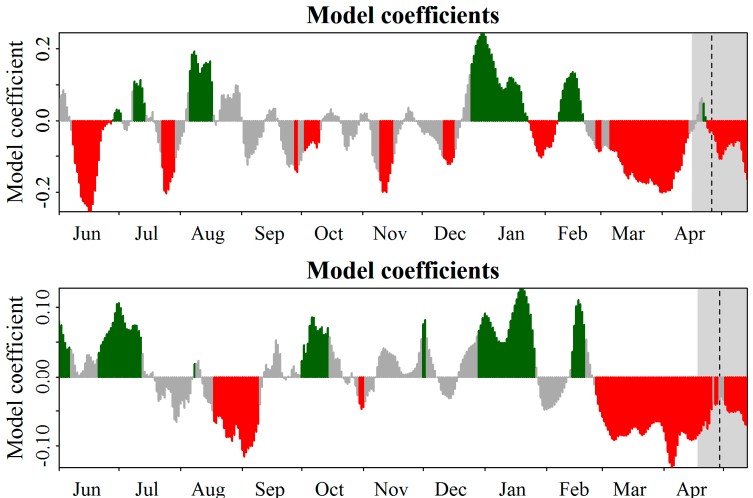

**Figure 4.** Delineating chilling and forcing phases of Picholine Marcaine cultivar cultivated at the INRA–Tassaout experimental station (Morocco), using the PLS approach by comparing a long series of flowering dates and temperatures during 29 years from 1986 to 2019 (**top panel**); with data of 'Picholine Marocaine' in WOGBM obtained for a sub-series of 6 years (2014–2019) (**bottom panel**).

**Table 2.** Chill and heat requirements of 'Picholine Marocaine' cultivar, estimated for two periods from 1986 to 2019 (29 years) and from 2014 to 2019 (6 years). The chill accumulation was estimated using three models; Chilling Hours Model, Utah Model and the Dynamic Model (mean ± standard deviation). The heat accumulation was estimated using the Growing Degree Hours Model (GDH). CV% is the coefficient of variation of each model over years for the period (1985–2019). CH: Chill Hour; CU: Chill Unit; CP: Chill Portion.

| | Chilling Phase | | | Forcing Phase | | | Chilling Hours Model | | Utah Model | | Dynamic Model | | GDH Model | |
|---|---|---|---|---|---|---|---|---|---|---|---|---|---|---|
| **Time Serie** | **Start** | **End** | **Duration** | **Start** | **End** | **Duration** | **CH** | **CV%** | **CU** | **CV%** | **CP** | **CV%** | **GDH** | **CV%** |
| 29 years (1986–2019) | 27 December | 21 February | 58 | 24 February | 16 May | 82 | 438 ± 114 | 25.94 | 512 ± 157 | 30.63 | 27.99 ± 6.16 | 22.02 | 21,559 ± 1952 | 9.05 |
| 6 years (2014–2019) | 30 December | 21 February | 53 | 27 February | 13 May | 76 | 459 ± 99 | ___ | 516 ± 150 | ___ | 26.25 ± 5.93 | ___ | 19,684 ± 1348 | ___ |

To explain the similar pattern between the two contrasting terms records (29 years versus 6 years), PLS outputs obtained from two identical short-term records of 6 years chosen among the 29 years with contrasting variability of winter temperature and flowering date were compared. Results of delineating of chilling and forcing accumulation phases based on a low variability of winter temperature and flowering was clearly distinct from the pattern based on long-term records, considered, here, as the reference. By using data with the highest variability in winter temperature and flowering data, a similar pattern as the reference was observed for both chill and heat phases (Figure S7). These results showed the importance of the inter-annual variability of winter temperature and FFD rather than long-term records to accurately delineate the chilling and forcing phases.

### 3.5. Comparison of Chilling Models

Using a long-term flowering record of 'Picholine Marocaine' cultivar (29 years), we compared the variation assessment of chill requirement obtained by the three most widely used models. The estimated average of winter chill for the 'Picholine Marocaine' cultivar during the chilling phase from 27 December to 23 February was $438 \pm 114$ Chill Hours, $512 \pm 157$ Chill Units and $27.99 \pm 6.16$ Chill Portions according, respectively, to the Chilling Hours, Utah and Dynamic models (Table 2). The coefficient of variation of chills calculated for the Dynamic model showed the lowest value (22.02%; Table 2) among the three models. Based on the lowest standard deviation criteria, the Dynamic model was chosen as the most accurate to assess olive chill requirements.

Using the single used GDH model for computing olive heat requirements, we found $21,559 \pm 1952$ GDH during the period from 24 February to 16 May (Table 2). The coefficient of variation of this period was low, 9.05%, indicating that the GDH is an accurate model.

### 3.6. Determination of Chill and Heat Requirements of WOGBM Cultivars

We used the Dynamic and GDH models to assess the chill and heat requirements of each of 285 cultivars of the WOGBM based on their delineated chilling and forcing phases obtained by PLS approach. Chill requirements among the cultivars showed high variability with low chill requirements' cultivars, with an average of $12.39 \pm 0.78$ CP (ranging from 10 to 13.83 CP), to high chill requirements' cultivars, with an average of $38.85 \pm 2.47$ CP (ranging from 36 CP to 43.67 CP) (Table 3; Figure S8).

**Table 3.** Clusters of chill requirements defined by NbClust function using Euclidean distance for dissimilarity measures and Ward's method for aggregation. Chill and heat requirements were calculated using the Dynamic and GDH models, respectively. DOY is day of the year, FFD means full flowering date and SD is standard deviation from the mean.

| Cluster of CR | Representative Cultivar * | Number of Cultivars | Chill Requirements | Heat Requirements | FFD (DOY) | | |
|---|---|---|---|---|---|---|---|
| | | | Mean $\pm$ SD | Mean $\pm$ SD | Mean $\pm$ SD | Min | Max |
| Low | Arbequina | 27 | $12.39 \pm 0.78$ | $20,467.17 \pm 2013.37$ | $122 \pm 3.09$ | 115 | 127 |
| Medium | Picholine Marocaine | 198 | $25.04 \pm 1.26$ | $20,882.92 \pm 747.82$ | $123 \pm 2.23$ | 115 | 130 |
| High | Rossello | 21 | $31.42 \pm 0.34$ | $21,382.66 \pm 1224.78$ | $124 \pm 3.22$ | 117 | 130 |
| Highest | Leccino | 39 | $38.85 \pm 2.47$ | $21,056.52 \pm 1026.91$ | $125 \pm 2.87$ | 118 | 132 |

* 'Arbequina' (Spanish cultivar largely cultivated in the world); 'Picholine Marocaine' (single dominant cultivar in Morocco); 'Rossello' (local Italian cultivar present in Tuscany); 'Leccino' (main Italian cultivar) [30].

Cultivars were classified into four distinct chill-requirements cultivar groups: low, medium, high and highest chill requirements cultivars (Figure 5a,b and Table S7). However, heat requirements did not reveal clear differences between cultivars. For instance, 98% of cultivars were between 19,088 and 22,644 GDH, except for six cultivars that had low heat requirements, which were ranged between 12,320 GDH and 18,691 GDH (Table S8).

### 3.7. Relationships between Flowering Dates and Chill and Heat Requirements

The amounts of chill requirements ranged between 10 and 44 CP depending on the cultivars (Table S8). Early flowering cultivars displayed low chill requirements with a significant correlation of about 33% (Figure 6), whereas the late flowering cultivars showed high chill requirements and low heat requirements. We observed a significant correlation between heat requirements and FFD (36%; Figure S9). These results indicated that the flowering date is governed by an interaction effect of chill and heat requirements.

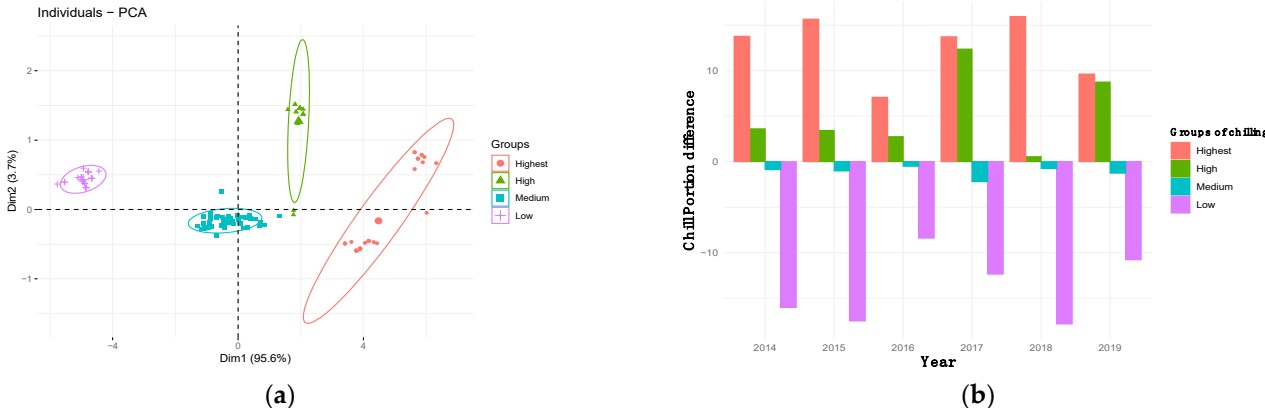

(**a**)                                                                                                       (**b**)

**Figure 5.** Graphical representation of the PCA (Principal Components Analysis) showing cultivars' groups according to their chill requirements calculated using the Dynamic model each year during the six years over the period 2014–2019 (**a**). Cultivars were clustered using the Euclidean distance and Ward's method for dissimilarity measures and agglomeration, respectively. The best number of chill clusters identified four groups according to the majority rule proposed by Charrad et al. [37] (Table S7). The graph at the right (**b**) represents the difference of variation for chill requirements among the cultivar groups in each year in comparison to the average calculated during the period 2014–2019.

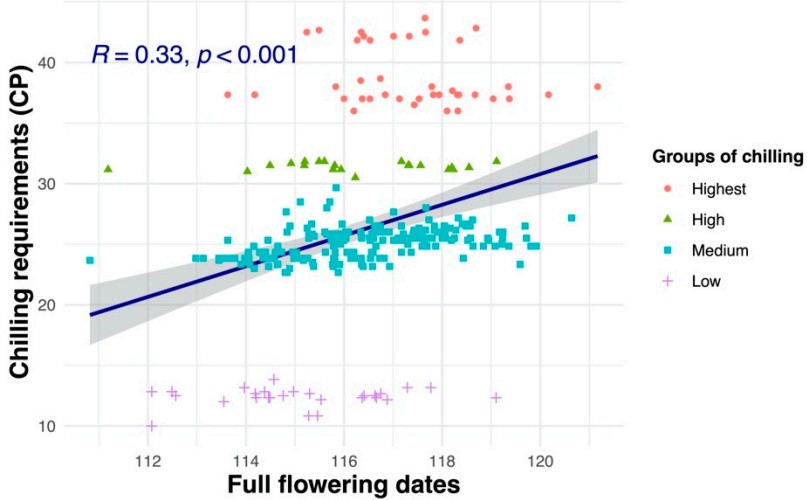

**Figure 6.** Correlation between full flowering dates of the four olive groups and their chill requirements (Chill Portions) according to the Dynamic model. $p < 0.001$ statistically significant.

To investigate the impacts of chilling and forcing temperatures on the flowering date, and which temperatures have more effect on the olive flowering process (winter chill or spring heat temperatures), the full flowering dates were plotted against the average of mean temperatures during both the chilling and the forcing periods as obtained by PLS analysis, using the Kriging interpolation method (Figure 7) [47,48].

We used the three-dimensional plot to represent the variation of flowering date with regards to temperature variation during the chilling and forcing phases previously delineated by PLS regression for the 'Picholine Marocaine' cultivar. The isolines displayed a clear slope pattern indicating that the flowering date for the 'Picholine Marocaine' cultivar is governed by both chill and heat requirements (Figure 7). Interestingly, we observed the same pattern based on long- and short-term records, similarly to delineating phases of chill and heat accumulation (Figure 7).

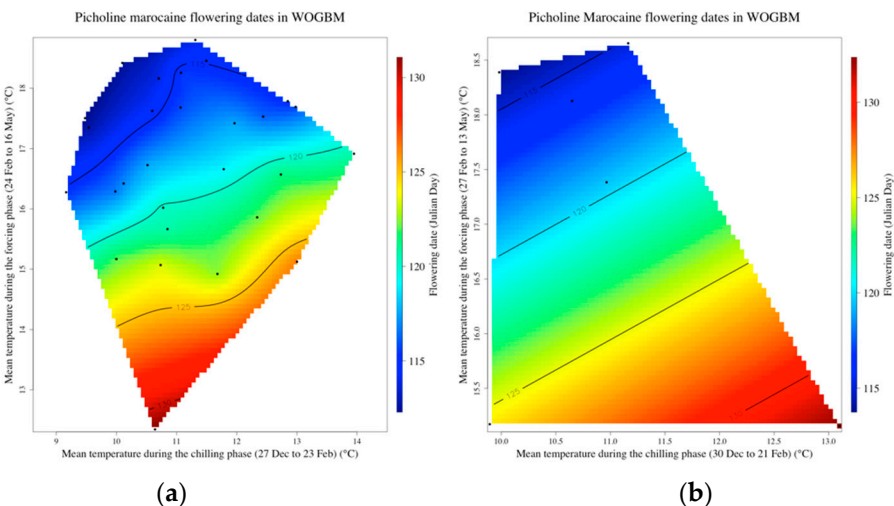

(**a**)　　　　　　　　　　　　　　　　　　　　　(**b**)

**Figure 7.** Response of 'Picholine Marocaine' full flowering dates to mean temperatures during the chilling and forcing phases for two periods (**a**) 1986–2019 and (**b**) 2014–2019. When isolines are parallel to the *x*-axis, changes in flowering date are clearly influenced by a variation in mean temperature during the forcing phase. However, isolines with clear slopes show that a variation in mean temperature during the chilling phase is responsible for flowering date time variation. The color spectrum must be regarded as a change in full flowering dates.

The Kriging interpolation method showed three patterns of the effect of temperature during the chilling and the forcing phases for all cultivars. Most of the cultivars (75%) displayed a similar pattern to the 'Picholine Marocaine' cultivar in which the temperature during both chilling and forcing phases influenced the full flowering date. For the remaining cultivars, the flowering date seems to be governed more by the forcing phase (17%) or by the chilling phase (only 8%; Figure 8 and Figure S10). Among 65 early bloomers, the flowering dates of 13 cultivars was governed much more by temperature during the chilling phase.

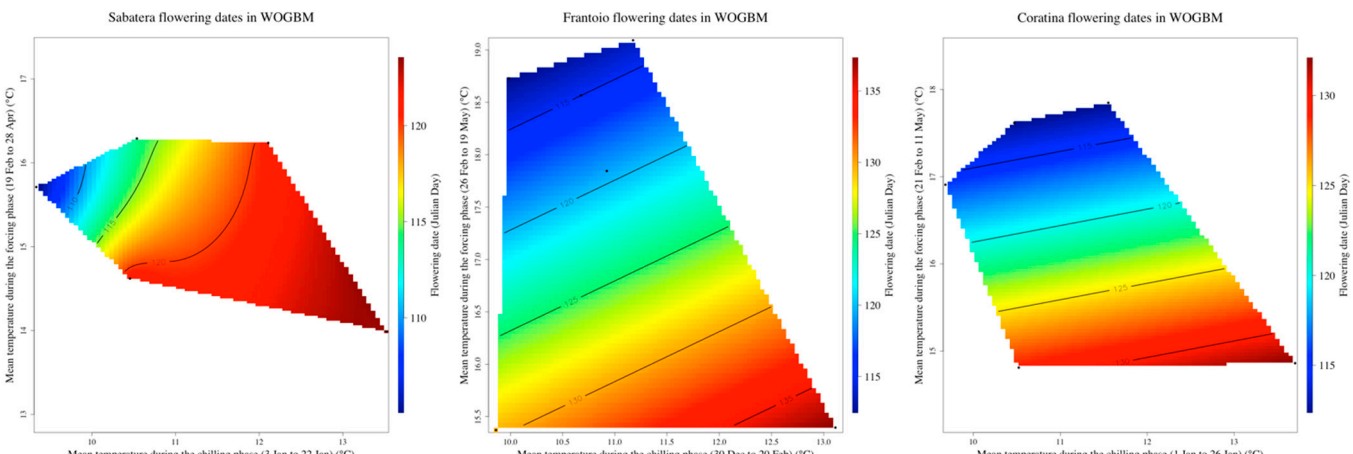

**Figure 8.** Different patterns of the effect of the average of mean temperature during the chilling and the forcing periods delineated by PLS analysis on cultivars' flowering dates. The three panels show the effect of the average of mean temperature during chilling period on the flowering date of 'Sabatera' (at the left) cultivar as well as 8% of analyzed cultivars, 'Frantoio' (in the middle), which is the case for 75% of analyzed cultivars (see Figure S10) and 'Coratina' (with horizontal isolines in the *x*-axis) representing the remaining cultivars.

## 4. Discussion

Perennial fruit species enter a dormancy phase in late autumn to avoid the freezing temperatures during winter in their respective habitat of origin [49]. Following the winter dormancy classification into endo- and ecodormancy, researchers have defined the concepts of chill and heat requirements, CR and HR, respectively, to represent the climatic needs of buds for overcoming the dormant state [50]. CR and HR are considered as two main driver factors for the timing of budburst and bloom in temperate tree species [51]. In olive trees, while clear relationships between winter cold temperature and flowering date were evidenced [17,19], links between CR and HR on flowering date are still misunderstood [52]. Several investigations into modelling the long term-record of airborne pollen data point out the importance of CR and HR as the main factors driving the olive flowering date [9,53]. Aiming to predict the flowering date for olive orchards, all those previously studied focused on large olive-growing areas with a single cultivar by using models that are designed for predictive investigations [54]. Here, we proposed, for the first time, investigations on a large set of olive genetic resources, 331 cultivars of the worldwide collection of Marrakech, to examine the variability in the observed flowering date, defined CR and HR accumulation phases and assessed CR and HR amounts. We discuss our results with regard to the following topics: (i) the importance of inter-annual variability of daily temperature to delineate CR and HR accumulation phases by the PLS approach; (ii) the selection of chill model and the accuracy of CR and HR assessment and (iii) the importance of CR and HR as driving factors for the flowering date.

### 4.1. Inter-Annual Variability of Daily Temperature Is More Important Than a Long-Term Record for the Accurate Delineation of Chill and Heat Accumulation Phases

The approach of Partial Least Squares Regression (PLS) has been shown over the last decade to be efficient and useful in determining the relevant phases involved in the flowering process of many species, including walnut [26,55], cherry trees [25], chestnut and jujube, almond and olive [4,23,27,56,57], apricot [58] and apple [59]. All these studies used long-term records of budburst or bloom data (≥15 years for one geographical site), while our data were collected on a short-term of 6 years (from 2014 to 2019). Before applying the PLS approach on the variability of 331 olive cultivars, we were concerned about whether a short-term record is sufficient to accurately delineate accumulation of the CR and HR phases. We used a long-term record of 29 years collected on 'Picholine Marocaine' cultivar in the Tassaout experimental site of the worldwide collection of Marrakech. This site is experiencing a global warming since there is a temperature increase of 0.35 (mean temperature), 0.41 (maximum) and 0.28 °C (minimum) on average for the five decades measured (see the Results section). El Yaccoubi et al. [4] evidenced the advanced flowering date pattern obtained on long-term record of both airborne pollen and phenology stage. As the advanced flowering date is a common pattern observed for several fruit trees such as almond and cherry [22,56,60,61] under global warming, we considered the long-term record of 29 years on 'Picholine Marocaine' as the reference model for our following investigations.

By using the PLS approach and taking into account the consensus on chill availability assessment by the three commonly used models (see the Results section), we considered the chilling phase of 'Picholine Marocaine' cultivar in the experimental station of Tassaout, Morocco, as delineated from 27 December to 23 February. Otherwise, we considered the phase of heat accumulation of 'Picholine Marocaine' cultivar as delineated from 24 February to the flowering date averaged to 16 May. For the delineated chilling accumulation phase, we noted a short period of 15 days of stable negative coefficients, which could be interpreted as forcing phase. However, this period fits well with the chill availability phase previously defined by a consensus of all three models. Further, we noted no significant effect (VIP < 0.8) for this period when PLS inputs were limited to the short-term records of 6 years. This gap could be explained by the unclear beginnings and ends of phases and the transitional phases between chilling and forcing [26]. While the exact start and end dates of chill and heat accumulation have been difficult to determine since they relied on experimentation

or were based on the researchers' subjective judgment, delineating these potential phases based solely on statistics became useful using the PLS approach as we obtained here and according to previous investigations on several fruit tree species [22,25,56,57,62,63]. However, all previous investigations were based on long-term records (>15 years for one geographical site), as we used for the 'Picholine Marocaine' cultivar with long-term records of 29 years.

Comparing the PLS outputs based on the long-term (29 years) and short-term records (6 years), we noted a similar pattern of delineating chill and heat accumulation phases. We hypothesized that this similar pattern is due to the inter-annual variability of daily temperatures within this short period. First, the variances in the inter-annual variability in the long (29 years) and the short period (6 years) were not significantly different. Second, we obtained delineated chill and heat accumulation phases similar to the reference based on long-term records, when we used data inputs with the highest variability of winter temperature and flowering dates. Hence, we concluded that the accuracy of delineating chill and heat accumulation phases is more sensitive to variability of inter-annual daily temperatures than to long-term records. These findings were supported by recent experimental investigations on apples and pears [64]. In a two-year experiment, the authors generated long-term phenology data by exposing potted trees to distinct environments during winter (records of tree phenology and hourly temperature for 66 and 32 experimental seasons in apples and pears, respectively). By using a PLS approach, Fernandez et al. [64] concluded that inter-annual variation may be more important than the number of seasons for delineating chill and forcing phases and estimating the cultivars' requirements.

Due to its inter-annual variation, the short-term record of 6 years corresponding to the period of phenology records of the worldwide collection seems to be suitable for the accurate delineation of chill and heat phases and assessment of the cultivars' requirements. However, before delineating potential phases and assessing CR and HR, we had to select the best suitable model.

### 4.2. Dynamic Model Is the Best Suitable to Accurately Assess CR for Cultivars of the Worldwide Collection of Marrakech

The most widely used chill models in horticulture are the Chilling Hours model (CHM) [41,42], the Utah model (UM) [43] and the Dynamic model (DM) [44,45]. The CHM simply calculates the number of hours when the temperature (T) is below 7.2 °C [41,42]. At a given time during the dormancy period (in hours after a fixed starting time at the beginning of the dormancy season), the number of accumulated Chill Hours (CH) is thus computed between 0 and 7.2 °C. This basic model lacks a mechanism to reduce the number of accumulated chill units when high temperatures occur. The chilling efficiency varies with temperature, including negative chill accumulation by high temperatures [43]. This mechanism was taken into account in the UM which established a weight function for different temperature intervals, implying differences between these intervals regarding their effectiveness in contributing to endodormancy release [43]. The most frequently used model in warm regions, and specifically in the Mediterranean climate areas, is the DM, which relies on the assumption that winter chill accumulates in a two-step process. Initially, cold temperatures stimulate the production of a chill precursor compound. Once a certain quantity of this intermediate has accumulated, it can be transformed into a so-called Chill Portion by a process requiring relatively warm temperatures [44,45]. These chill models differ greatly in their accuracy, with severe implications for the validity of CR estimates from one location in other growing regions [65]. For these reasons, we opted to compare the accuracy of these models to select the best one that was suitable to assess CR for all cultivars. According to the assumption that the CR is genetically predetermined, we used the lowest coefficient of variation as selective criteria for the best accurate model. Chill amounts computed between 27 December and 23 February varied over years for the three models, but were much more clearly defined for the DM than for the other two models, according to coefficients of variation 22.02% for DM, 25.94% for CHM and 30.63% for UM.

These findings support previous investigations showing the DM to perform equally well or better than the other models [8,25,66]. Interestingly, we obtained similar chill amounts for the 'Picholine Marocaine' cultivar based on long-term and short-term records, indicating that DM accuracy is consistent with the assessment of chill amount for each of cultivars based on a short-term record of 6 years.

*4.3. CR Patterns and Their Possible Significance to Adaptive Traits of Cultivars*

According to their delineated chill phase and using the DM, we computed CR for each of the 285 cultivars and we classified the variability ranging from 10 to 43.7 CP into four CR patterns, low (27 cultivars), medium (198), high (21) and highest CR pattern (39). Interestingly, we noted a tendency of early bloomers with low CR as shown by a significant correlation (R = 0.33, P = 1.2 $\times 10^{-8}$). Diez-Palet et al. [22] reported similar findings on almond and apple cultivars with a higher correlation than in our study (>0.55). However, these authors used flowering day (DOY) while we computed the correlation based on BLUPs to investigate the genetic effect of flowering date. The range variation of BLUP values is limited compared to those of flowering day. Hence, we evidenced a similar pattern reported in almond and apple, although some early bloomers can have a high CR but some of these cultivars have a low HR (see Figure S9). Among cultivars with a low CR, 37% were early bloomers while they represent 21.7% among all 285 studied cultivars (Table S8). Otherwise, we noted 66.7% of the 39 cultivars of the highest CR were delayed bloomers. These observations support the pattern of early bloomers with low chill and late bloomers with high chill, as reported in previous investigations [8,22,56].

Here, we are aware that our CR assessment is totally linked to the accurate delineating of chill phase for each cultivar. Considering solely one assessed CR value as a specific trait for a given cultivar is likely the wrong approach; meanwhile our results have to be considered as a classification grid. For a given cultivar, its CR assessment is useful when it is compared to other cultivars of its pattern and belonging to others. For instance, Arbequina was classified as an early bloomer cultivar with a low CR in the same way as 'Chalikidikis' (from Greece), 'Coratina' (Italy) and 'Farga' (Spain); meanwhile, 'Leccino' was classified as mid bloomer cultivar with the highest CR, such as 'Allora', 'Morchione' and 'Rossellino Cerretano' from Tuscany (Italy; Table S8).

As the majority of the 285 studied cultivars (69.5%) had medium CR, we considered the two opposite patterns, low (27 cultivars) and highest CR (39), as the most interesting for investigating whether their significance was linked to a possible adaptive trait. According to the pattern we evidenced above indicating early bloomers with low CR and delayed bloomers with the highest CR, we investigated the links between the geographical origin of cultivars, when it was known, and their pattern. We hypothesize that cultivars from south shore of Mediterranean have a low CR while those from the north shore have the highest CR. Among the 10 early bloomers with low CR, the cultivars were from Egypt (1) and Tunisia (2), the remaining were from Spain (5), such as Arbequina, 'Farga' and 'Sabatera', from south Italy, 'Coratina', and Greece, 'Chalkidikis'. There are no clear links between the origin of cultivars and their pattern as they are issued from areas not limited to the north or south shore of Mediterranean. Interestingly, 16 out of 26 (61.5%) delayed bloomers cultivars originated from the north shore of Mediterranean, such as 'Allora', 'Morchione' and 'Rossellino Cerretano' from Tuscany (Italy; Table S8). These results support our assumption on the links between the geographic origin and the CR pattern. More investigations focusing on these two contrasting patterns, early bloomers with low CR versus delayed bloomers with the highest CR, will give insights on the relationships between flowering date and CR and their significance to a possible adaptive trait.

*4.4. HR Variability and Limits of Its Assessment*

Here, as CR, we computed HR based on the phase we delineated by a PLS approach and using the ChillR package [32]. However, we computed CR and HR without any assumption about the relationship between both phases, and in particular, we did not

assume a sequential [13] nor an overlap model [67]. We focused on delineating chill and heat phases by using the PLS approach on temperature records that have influenced the flowering date without taking into account a possible interaction between cold and hot temperatures. Hence, we defined the HR phase as starting in the day following the end date of the chill phase, while knowing that cold and hot temperatures are likely concurrently driving the dormancy release and flowering date. Chuine et al. [68] demonstrated that models not calibrated with endodormancy break dates can generate large discrepancies in forecasted budbreak or flowering dates when using climate scenarios as compared to models calibrated with endodormancy break dates.

For these reasons, our results on HR assessment have to be taken with caution. Looking for HR variability, we did not reveal clear differences between cultivars since 98% of them displayed a limited range variation from 19,088 to 22,644 GDH. However, we observed a significant correlation between flowering date and HR amount indicating a pattern tendency of early bloomers with low HR. The same results obtained by [9,69] confirm the relationship between heat accumulation and flowering date.

### 4.5. Flowering Date of Olive Tree Is Likely Governed by Cold and Heat Temperatures

According to delineation of the chill and heat requirements phases obtained by PLS approach, flowering date is likely governed by an interaction effect of chill and heat requirements. Rojo et al. [9] reported that chill and heat accumulations were related to the onset of the flowering period. Thus, early flowering cultivars displayed low chill requirements with a significant correlation of about 33% (Figure 6). However, late flowering cultivars showed high chill requirements but low heat requirements. We observed a significant correlation between heat requirements and flowering date (36%; Figure S9). These results indicated that flowering date is governed by an interaction effect of chill and heat requirements. Our results are consistent with previous studies on pear and peach trees [70–72].

### 5. Conclusions

Our results concerning the flowering dates of 331 cultivars belonging to the WOGBM showed an important significant year effect, followed by cultivar and an interaction effect on FFD. These cultivars, representing Mediterranean olive genetic variability, are classified into four groups of FFD: the early; mid; late and extra late flowering groups, with an average of flowering dates of DOY 119 (29 April), 123 (3 May), 125 (5 May), 128 (8 May), respectively.

This study is a detailed investigation of the thermal requirements of cultivated olive. Different CR and HR were obtained for 285 cultivars in this study. The wide variation in CR between cultivars suggests that early bloomers typically have a lower CR. There were no observable differences between cultivars according to HR variability. We did notice a pattern of early bloomers with a low HR, though. The FFD has proven to be likely governed by an interaction effect of CR and HR. Therefore, further investigations are necessary to address the issue of lower CR and higher HR cultivars under future climate scenarios.

Future research on the effects of climate change on the phenology of cultivated olive in the Mediterranean region may benefit from our findings. Therefore, this study will be useful for developing climate adaptation strategies that guarantee CR fulfillment of olive cultivars in a context of global warming. Our classification of olive Mediterranean genetic resources should be validated by future research, combining the statistical approaches with experimental approaches.

**Supplementary Materials:** The following supporting information can be downloaded at: https://www.mdpi.com/article/10.3390/agronomy12122975/s1, Figure S1: Daily temperature variation over a year computed on mean, maximum and minimum temperature data collected from 1972 to 2019. For each metric, the colored area indicates the variation over 47 years in the experimental station of Tassaout. Min temperature below 7.2 °C is approximatively occurring during 120 days starting from mid-November to the beginning of march, a suitable period of chilling requirement of olive tree (see Figure 3 and Results); Figure S2: Annual temperature trends (mean, maximum and minimum) in

the worldwide collection of Marrakech (WOGBM) during the period 1972–2019. Global warming is clearly shown since there is a temperature increase of 0.35 (mean temperature), 0.41 (maximum) and 0.28 °C (minimum) on average for each decade; Figure S3: Rainfall variation over 47 years from 1972 to 2019 in the experimental station of Tassaout; Figure S4: Correlation between phenological stages from bud burst (stage 51) to the end of flowering (stage 69) and the flowering duration for the 331 cultivars of the worldwide collection Marrakech over six years 2014–2019 investigated by Pearson correlation analysis A significant correlation is evidenced between phenological stages related to the inflorescence emergence (stages 51, 54 and 55) and to flowering (stages 61, 65 and 69). Strikingly, the budburst stage 51 is significantly negatively correlated to the flowering duration measured as the DOY (day of the year) difference between the beginning of flowering (stage 61) to the end of flowering (69). Interestingly, blooming time (stage 65) is significantly correlated to all phenological stages including those related to the inflorescence emergence; Figure S5: Boxplots of full flowering dates observed in the cultivars of the worldwide collection of Marrakech (WOGBM). The first and the third quartiles are represented by both sides of the box and the whiskers' tips represent the standard deviation of the mean value. The line in the middle is always the median. We clearly note a huge interannual variability of blooming time in one geographical site (the experimental station of Tassaout) likely due to interannual temperature variation described in the Figures S1 and S2; Figure S6: Assessment of chill availability (The beginning to the end of chilling accumulation) using the most widely used chill models, the Chilling Hours model (CHM), the Utah model (UM) and the Dynamic model (DM). Based on 15 days running mean of daily temperatures in the experimental station of Tassaout, Morocco, over 47 years from 1972 to 2019, chill availability was assessed using the ChillR package according to CHM and DM models (Figure S6a) and UTAH model (Figure S6b). Whatever the model used, chill is available during a period of 152 days starting from first week of November to the end of March and the beginning of April; Figure S7: Delineation of the chilling and forcing phases by the PLS approach for two series of recorded data for 6 seasons. (A) seasons with the lowest variability for chilling and blooming dates, (B) seasons with the highest variability of chilling and blooming dates; Figure S8: Chilling and forcing phases by the PLS approach of four different cultivars representative of four groups of chilling (see table S6); Figure S9: Correlation between olive cultivar effect of full flowering time and the heat requirements (GDH). $p < 0.001$ Statistically significant; Figure S10: Percentage of patterns of the effect of temperatures on flowering dates of olive cultivars; Table S1: List of 331 studied olive cultivars, identified using 20 SSR markers and 11 endocarp traits by El Bakkali et al. [30]. The number of trees, cultivar's origin of collection, accession code, accession name and cultivars for which PLS analysis was completed are mentioned. (Excel file); Table S2: Analysis of variance of full flowering dates (Stage 65 according to BBCH scale of olive, Sanz Cortés et al., 2002) as a function of cultivar, year and cultivar * year interaction. Alpha is 0.05. Df is the degree of freedom. Levels of significance: ns (not significant); * ($p<0.05$); ** ($p<0.01$); *** ($p<0.001$); Table S3: Analysis of variance and Tukey test mean comparison of cultivar's full flowering dates. Means with different letters are significantly different at Alpha of 5%. (Excel file); Table S4: Best number of clusters of cultivars by their full flowering dates according to the results of eight indices. Dissimilarity measures were calculated between cultivars by Euclidean distance, and agglomeration was executed by Ward's method (1963). Four clusters were chosen to be the best number of clusters according to the majority rule by Charrad et al. [37]. In this case, four out of eight indices proposed four as the best number of clusters. For more information about NbClust function in R and indices computation, see the publication of Charrad et al. [37] (full reference in main document); Table S5: Analysis of variance and Tukey test of four groups of full flowering dates suggested by NbClust function in R [37] and obtained by Hierarchical clustering using Euclidean distance for dissimilarity measures and Ward's method for agglomeration. FFD is Full flowering date. Alpha is 0.05; Table S6: Best number of clusters of cultivars by their BLUP of full flowering dates according to results of eight indices. Dissimilarity measures were calculated between cultivars by Euclidean distance, and agglomeration was executed by Ward's method (1963). Four clusters were chosen to be the best number of clusters according to the majority rule by Charrad et al. [37]. In this case four out of eight indices proposed four as the best number of clusters; Table S7: Best number of clusters of cultivars by their chilling requirements (See Materials and Methods section) according to the results of twenty-six indices. Dissimilarity measures were calculated between cultivars by Euclidean distance, and agglomeration was executed by Ward's method (1963). Four clusters were chosen to be the best number of clusters according to the majority rule by Charrad et al. [37]. In this case seven out of twenty-six indices proposed four as the best number of clusters; Table S8: The

chilling and heat requirements of analyzed WOGBM cultivars. Delineated by the PLS approach and estimated by the dynamic model (DM) for chilling and GDH model for forcing. FFD is Full flowering date, BLUP is Best linear unbiased prediction, CR is chilling requirement, HR is heat requirement and Std is standard deviation. (Excel file).

**Author Contributions:** Conceptualization, B.K., H.Z., A.E.Y. and A.M.; methodology, B.K., O.A.-S., H.Z., A.E.Y. and A.M.; software, O.A.-S. and B.K.; validation, all authors.; formal analysis, O.A.-S. and B.K.; investigation, all authors; data curation, O.A.-S., A.E.Y., A.M., A.E.B., H.Z. and B.K.; writing—original draft preparation, O.A.-S. and B.K.; writing—review and editing, all authors.; visualization, O.A.-S. and B.K.; supervision, B.K. and H.Z.; project administration, B.K.; funding acquisition, B.K., C.E.M. and A.E.B. All authors have read and agreed to the published version of the manuscript.

**Funding:** This research was funded through Labex AGRO 2011-LABX-002, project N° 2003-001 "ClimOliveMed" (under I-Site Muse framework) coordinated by Agropolis Fondation.

**Institutional Review Board Statement:** Not applicable.

**Informed Consent Statement:** Informed consent was obtained from all subjects involved in the study.

**Data Availability Statement:** Not applicable.

**Acknowledgments:** We warmly thank Benoit Pallas for helpful analysis data and scientific discussions, the INRA Morocco direction staff for supporting all phenotyping work and the EVOLEA project for supporting the PhD student, O. A-S, during his training at AGAP Institute Montpellier. We also thank the Hassan II Academy of Sciences and Technology and the Ministry of Higher Education, Scientific Research and Innovation (Morocco) for supporting Moroccan co-authors as part of ClimGenOlive project. Adnane El Yaacoubi was supported by the PRIMA program (AdaMedOr project, 2020-2023) funded by the Ministry of Higher Education, Scientific Research and Innovation.

**Conflicts of Interest:** The authors declare no conflict of interest. The funders had no role in the design of the study; in the collection, analyses, or interpretation of data; in the writing of the manuscript, or in the decision to publish the results.

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
