# Peer review of "Statistical Approach to Assess Chill and Heat Requirements of Olive Tree Based on Flowering Date and Temperatures Data: Towards Selection of Adapted Cultivars to Global Warming"

_agronomy, doi:10.3390/agronomy12122975_

Round 1

Reviewer 1 Report

Manuscript evaluates the chill and heat requirements of olive tree, using flowering data collected over six years on 331 cultivars cultivated in the collection of Marrakech, Morocco. The authors classified olive cultivars into four groups according to their chill requirements. The paper is generally well structured and presented.

Materials and methods are appropriate. The obtained results are statistically interpreted and discussed in relation to the specialized literature.

The study present scientific and practical information for producers of olives in Mediterranean region.

However, there are some shortcomings:

In the abstract section, specify why the obtained results are important.

 Include soil information from researched experiences

-        Include some description for representative cultivars of olive

-        For all equipment and software, at least the company, model and country of origin must be mentioned.

Reviewer 2 Report

Brief summary

The paper focuses on the relationships between phenology and air temperature of 285 olive cultivars. The statistical analysis was based on the 6- years series (2014-2019) of phenological data collected at Marrakech (Morocco) and the temperatures data provided by a local weather station. A preliminary analysis of a 29-year time-series for one cultivar (Picholine Marocaine) showed that the results derived from the full series were consistent with those obtained with the 2014-2019 subset of this series and that the variability of inter-annual temperatures is the most important factor in determining chilling and forcing periods.

The best linear unbiased prediction (BLUP) was calculated to extract the cultivar effect on the full flowering date. The chilling and heat requirements were estimated for all the 285 cultivars, which were classified in 4 groups, according with their different chill requirements. A clear difference in chill requirements was observed among the groups, while heat requirements were less variable among cultivars.

General comments

The work presented in this manuscript falls within the scope of the Journal and is very relevant to climate change adaptation, as it offers a comprehensive analysis of the relationships between air temperature and olive phenology, by considering a large number of cultivars. The approach is innovative, because the study analyzes several cultivars together and is based on their short-term (6 years) phenological series (in other studies series of at least 15 years are used); this choice is based on a statistical analysis and is well argued.

The manuscript is well presented, with an appropriate title and an informative abstract; the writing is logical and quite easy to comprehend. The Introduction is effective in contextualizing the study and defining its objectives. The Material and methods section is very well structured and provides a clear rationale for the specific methodological choices, although some further explanations could help readers in interpreting the model coefficients generated by the PLS analysis (see specific comments). The results are properly organized and clearly presented; some figures in supplementary materials should be moved to this section to improve readability (see specific comments). The discussion is clear and very focused on the most relevant results, which are adequately compared with the background of previous work; it also properly highlights the main limitations of the study.

Specific comments

Page 1, line 40: replace “It mainly involved” with “It is mainly involved”

Page 2, line 91: please, specify the CRS (coordinate reference system)

Page 3, line 121: please, specify the equation applied by the chillR package to derive the hourly data

Page 4, line 169: it is not clear what period the generated dataset of hourly temperatures covers (1972 to 2019, as reported here or 1972 to 2012, as reported at line 121?)

Page 5, line 190: “therefore” is repeated

Page 5, line 193- 194: the sentence from “Positive model….day delay bloom” is not clear. In addition, with the aim to improving readability, I suggest better explaining the signs of coefficients, as in the cited paper of Rojo et al. 2020 [9]: …positive coefficients imply that the temperatures (…) were positively correlated with the flowering dates, i.e., lower temperatures during winter lead to earlier flowering dates (equivalent to the chilling accumulation period). In contrast, negative coefficients signify that the temperatures were negatively correlated with the flowering dates, i.e., higher temperatures in spring also contribute to earlier flowering (equivalent to the forcing period)

Page 6, line 243: it is not Fig. S3: maybe it is Fig. S1

Page 12, line 421: please, consider moving figure S10 from Supplementary materials to the paper, it can improve the readability.

Page 13, line 444: replace “ovoid” with “avoid”

Page 13, line 451: replace Links with “links”

Page 21, line 900: this reference is duplicated (see [9])
